# Impact of Developmental Changes of GABA_A_ Receptors on Interneuron-NG2 Glia Transmission in the Hippocampus

**DOI:** 10.3390/ijms241713490

**Published:** 2023-08-30

**Authors:** Linda Patt, Dario Tascio, Catia Domingos, Aline Timmermann, Ronald Jabs, Christian Henneberger, Christian Steinhäuser, Gerald Seifert

**Affiliations:** 1Institute of Cellular Neurosciences, Medical Faculty, University of Bonn, 53127 Bonn, Germany; linda.patt@ukbonn.de (L.P.); dario.tascio@ukbonn.de (D.T.); ciafonsodomingos@gmail.com (C.D.); aline.timmermann@web.de (A.T.); ronald.jabs@ukbonn.de (R.J.); christian.henneberger@uni-bonn.de (C.H.); 2German Center for Neurodegenerative Diseases (DZNE), 53127 Bonn, Germany

**Keywords:** NG2 glia, GABA_A_ receptors, postnatal development, hippocampus, neuron–glia interaction, extrasynaptic receptors, gephyrin, radixin, expansion microscopy

## Abstract

NG2 glia receive synaptic input from neurons, but the functional impact of this glial innervation is not well understood. In the developing cerebellum and somatosensory cortex the GABAergic input might regulate NG2 glia differentiation and myelination, and a switch from synaptic to extrasynaptic neuron–glia signaling was reported in the latter region. Myelination in the hippocampus is sparse, and most NG2 glia retain their phenotype throughout adulthood, raising the question of the properties and function of neuron-NG2 glia synapses in that brain region. Here, we compared spontaneous and evoked GABA_A_ receptor-mediated currents of NG2 glia in juvenile and adult hippocampi of mice of either sex and assessed the mode of interneuron–glial signaling changes during development. With patch-clamp and pharmacological analyses, we found a decrease in innervation of hippocampal NG2 glia between postnatal days 10 and 60. At the adult stage, enhanced activation of extrasynaptic receptors occurred, indicating a spillover of GABA. This switch from synaptic to extrasynaptic receptor activation was accompanied by downregulation of γ2 and upregulation of the α5 subunit. Molecular analyses and high-resolution expansion microscopy revealed mechanisms of glial GABA_A_ receptor trafficking and clustering. We found that gephyrin and radixin are organized in separate clusters along glial processes. Surprisingly, the developmental loss of γ2 and postsynaptic receptors were not accompanied by altered glial expression of scaffolding proteins, auxiliary receptor subunits or postsynaptic interaction proteins. The GABAergic input to NG2 glia might contribute to the release of neurotrophic factors from these cells and influence neuronal synaptic plasticity.

## 1. Introduction

NG2 glia represent the largest population of proliferative cells in the postnatal brain outside the neurogenic niches. In white matter, the majority of these cells differentiate into oligodendrocytes and are therefore also named oligodendrocyte progenitor cells or OPCs [1]. In contrast, in the hippocampus and other gray matter regions, most of these cells do not become oligodendrocytes but maintain their NG2 phenotype throughout adulthood [2,3]. NG2 glia specifically express the proteoglycan NG2 (also termed CSPG4) and PDGFRα [4], which helps identifying these cells. They are ubiquitously distributed throughout the brain, with the cell density of astrocytes exceeding that of NG2 glia by up to 16 fold, depending on the brain region [5]. In the hippocampus, they form a relatively homogeneous population with regard to their shape, orientation and distribution across the different strata [6]. NG2 glia are equipped with functional AMPA and GABA_A_ receptors and receive direct synaptic input from glutamatergic and GABAergic neurons [7,8,9], but the impact of synaptic innervation of NG2 glia is not well understood.

GABA_A_ receptors are ionotropic receptors forming pentamers composed of 2α, 2β and 1γ subunits. Furthermore, 16 different GABA_A_ receptor subunits are known (6α, 3β, 3γ, δ, ε, π, θ), forming subunit combinations with distinct pharmacological properties. Pharmacological evidence of cell-specific subunit composition is derived from modulators acting on allosteric binding sites for benzodiazepine, barbiturate, zinc and volatile anesthetics (reviewed by [10,11,12]). The receptors form a Cl^−^- and HCO_3_^−^-permeable channel. In adult neurons, GABA_A_ receptor opening leads to hyperpolarization, while in the immature brain, the Cl^−^ equilibrium potential is more positive and GABA depolarizes the membrane [13]. In rodents, this developmental switch occurs during the first two postnatal weeks and is due to increasing expression of the KCl co-transporter KCC2 [14,15]. Oligodendrocytes and NG2 glia lack KCC2, and therefore, GABA_A_ receptor opening entails depolarization [16,17,18].

Previously, we have shown that at the juvenile stage, receptor responses of NG2 glia are potentiated by allosteric modulators interacting with the benzodiazepine and barbiturate binding sites. GABA-mediated currents were also potentiated by zolpidem, which binds at the α/γ2-benzodiazepine site. On a subcellular level, the γ2 subunit is located postsynaptically in NG2 glia while tonic currents are mediated by zolpidem-insensitive, extrasynaptic receptors lacking γ2 [17,19]. A developmental switch from synaptic to extrasynaptic GABAergic interneuron-to-NG2 glia signaling was observed in the barrel cortex, which was due to downregulation of the γ2 subunit [19,20]. 

In the present study, we compared spontaneous and evoked GABA_A_ receptor-mediated currents of NG2 glia in the juvenile and adult hippocampus and asked whether the mode of interneuron–glial signaling changes during postnatal development. Moreover, molecular analyses were performed to reveal mechanisms of glial GABA_A_ receptor trafficking and clustering. We found a significant decrease in synaptic innervation of hippocampal NG2 glia between postnatal days (p) 10 and 60. At the adult stage, enhanced activation of extrasynaptic receptors was observed, indicating a spillover of GABA. This switch from synaptic to extrasynaptic receptor activation was accompanied by a downregulation of γ2 and upregulation of the α5 subunit. 

In neurons, the γ2 subunit is an important interaction partner of the scaffolding protein gephyrin and crucial for postsynaptic GABA_A_ receptor clustering. These γ2-containing receptors form complexes with GARLH4 and neuroligin 2 [21,22]. For recruitment of gephyrin and maintenance of postsynaptic receptors, expression of collybistin, an interaction partner of neuroligin 2, is necessary [23,24]. In contrast, extrasynaptic α5 containing GABA_A_ receptors are anchored to the cytoskeleton by radixin [25,26]. The mechanisms underlying formation of receptor clusters in NG2 glia are not known. Here, we report that gephyrin and radixin are organized in separate clusters along glial processes. Surprisingly, despite the significant loss of γ2 and postsynaptic receptors with increasing postnatal age, in the glial cells, these changes were not accompanied by altered expression of scaffolding proteins, auxiliary receptor subunits or postsynaptic interaction proteins. 

## 2. Results

### 2.1. GABAergic Synaptic Input to Hippocampal NG2 Glia Declines during Maturation

We compared spontaneous synaptic currents (sPSCs) of NG2 glia in the CA1 stratum radiatum at p10 and p60 using a blocking solution containing NBQX (10 µM), D-APV (50 µM) and CGP55845 (5 µM) (original recordings shown in Figure 1A). We observed a significant reduction in sPSC frequency in cells of adult mice, and accordingly, the interevent interval between sPSCs was much longer at p60 (Figure 1B). In contrast, amplitude and kinetics of the sPSCs remained unchanged during development (Figure 1C,D). 

Next, miniature postsynaptic currents (mPSCs) were compared in the presence of the above blockers plus TTX (0.5 µM, recording time 5 min, Figure 2A,C). As for sPSCs, only recordings with more than two mPSCs/5 min were included into the analysis. mPSCs occurred less frequently than sPSCs, but their frequency also declined during development, and the interevent intervals became longer (Figure 2B). Peak amplitude, rise and decay time of mPSCs remained unchanged during maturation (Figure 2D). Together, the data indicate that in the hippocampal CA1 region, interneuron-to-NG2 glia signaling is downregulated during postnatal development, possibly due to diminished synaptic innervation. 

### 2.2. Pharmacological Analysis of Evoked Postsynaptic Currents (ePSCs)

We tested whether these functional changes were accompanied by alterations in postsynaptic receptor composition or anchoring. Given the low sPSCs frequency, for pharmacological analyses, we evoked PSCs (ePSCs) by near-field stimulation and assessed the efficiency of ligands for the benzodiazepine binding site changes during development (Figure 3). This site is located at the interface of α- and γ-subunits of GABA_A_ receptors [10]. Diazepam and zolpidem are γ subunit-specific modulators of GABA_A_ receptors. Receptors devoid of γ2 are less sensitive to diazepam, while low concentrations of zolpidem selectively activate α1/γ2 containing receptors [11,27]. In juvenile NG2 glia, diazepam enhanced ePSC amplitudes in five of seven cells while at p60, fewer cells were sensitive to the modulator (Figure 3(A1,A2)). Similarly, zolpidem potentiated ePSCs more frequently in juvenile NG2 glia (Figure 3(B1,B2)). Overall, the decreased sensitivity of GABA_A_ receptors to diazepam and zolpidem in adult NG2 glia might have indicated a developmental downregulation of the γ2 subunit.

Comparing the kinetics of ePSCs, we found that rise time and decay time slowed down with increasing age (Figure 3C,D). For rise time analysis, only recordings where stimulus artefacts were clearly separated from the ePSCs were included. Slower event kinetics could indicate that in NG2 glia of adult hippocampus, phasic GABAergic responses were also due to GABA spillover and activation of extrasynaptic receptors. 

### 2.3. Enhanced GABA Spillover and Extrasynaptic Receptor Activation in Adult NG2 Glia

To further pursue the hypothesis of spillover-mediated activation of NG2 glia at the adult stage, GABA transporters (GATs) were blocked. Two types of GATs are expressed in the mouse hippocampus, GAT-1 and GAT-3 [28,29], with the former being present in extrasynaptic membranes of presynaptic neurons while the latter has been mainly found in astrocytes. These transporters limit GABA diffusion and help keeping synaptic and extrasynaptic neurotransmitter levels constant [30]. We blocked GAT-1 and GAT-3 with NNC-711 and SNAP-5114, respectively, while evoking PSCs at interneuron-to-NG2 glia synapses in the CA1 stratum radiatum. At p10, neither NNC-711 nor SNAP-5114 had an effect on ePSC kinetics upon low-frequency, single pulse stimulation (Figure 4). In contrast, at p60, blocking GABA uptake significantly prolonged the decay (Figure 4). As expected, the effect of NNC-711 was stronger than that of SNAP-5114 (two sample *t*-test *p* < 0.01), reflecting the primary role of GAT-1 in regulating extrasynaptic GABA concentration due to its higher expression and more proximal localization to the synaptic cleft. These results are in line with the hypothesis that during later stages of development, NG2 glia detect GABAergic input predominantly through extrasynaptic receptors. 

### 2.4. GABAergic Input onto Adult Hippocampal NG2 Glia Is Mediated by α5-Containing Receptors

We further characterized pharmacological properties of extrasynaptic currents GABA_A_ receptors in NG2 glia. To increase ambient GABA concentration and unmask tonic currents, nipecotic acid, a broad spectrum GABA transporter inhibitor, was applied (1 mM, at least 3 min). Subsequent addition of the competitive GABA_A_ receptor antagonist, bicuculline (20 µM), induced a positive shift in holding current, which was due to block of GABA-mediated tonic currents (Figure 5A). The amplitudes of tonic currents did not change during development (Figure 5B). The α5 subunit of GABA_A_ receptors is mainly expressed extrasynaptically [31]. To test for its functional expression in NG2 glia, the α5-specific blocker α5IA (0.5 mM) was applied subsequent to nipecotic acid (1 mM, at least 3 min). Two min after α5IA, bicuculline (20 µM) was added to block remaining tonic currents (Figure 5C), and the effect of α5IA on the holding current was compared with the effect of α5IA + bicuculline. Interestingly, the inhibitory effect of α5IA was higher in NG2 glia of adult mice (Figure 5D), indicating that with ongoing development, an increasing proportion of α5-bearing receptors contribute to the activation of NG2 glia by extrasynaptic GABA. 

### 2.5. RT-PCR Substantiates Developmental Rearrangements of Glial GABA_A_ Receptors

To investigate putative structural changes accounting for the pharmacological changes of the receptors with increasing age, we isolated individual NG2 glia from the adult CA1 stratum radiatum, performed single-cell RT-PCR subsequent to their functional characterization (Figure 6A) and compared the data with the juvenile stage that we had previously analyzed [17]. Subunits β2 and β3 were differentiated using restriction analysis. The subunits α1, α2, α3, α4 and α5 subunits displayed relative expression frequencies of 0.5, 0.81, 0.56, 0.56 and 0.88 (n = 16) while those of β1, β2 and β3 were 0.5, 0.8 and 0.6 (n = 16). The frequency of γ1, γ2 and γ3 were 0.88, 0.19 and 0.31 (n = 17) while the δ subunit was only rarely expressed (expression frequency 0.08; n = 13) (Figure 6B, blue bars). Compared to the juvenile expression pattern (red bars in Figure 5B), the frequency of α5 and α3 expression were strongly upregulated at p60 while β3 and γ2 were less frequently found in cells from adult mice. This shift in transcript expression pattern correlated well with the functional changes of the receptors during development.

To further elaborate alterations of α5 and γ2 subunit expression with progressing age, we performed semiquantitative real-time RT-PCR with FAC-sorted NG2 glia. The normalized γ2 expression level was again reduced in adulthood (p10: 0.012 ± 0.009, n = 13; p60: 0.004 ± 0.004, n = 13; *p* < 0.05) while α5 remained unchanged (p10: 0.003 ± 0.002, n = 11; p60: 0.005 ± 0.005, n = 13; *p* = 0.27; *t*-test). 

### 2.6. GABA_A_ Receptor Clustering in Hippocampal NG2 Glia

We wondered whether the alterations of synaptic innervation and transcript expression were accompanied by changes in clustering of GABA_A_ receptors in developing NG2 glia. Structurally, synapses in NG2 glia look similar to neuron–neuron synapses [32], but formation of GABA_A_ receptor clusters at glial postsynapses has not yet been investigated. Passlick et al. [33] reported that NG2 glia express neuroligin-2, which in neurons is important for formation and maturation of inhibitory receptors. Interacting with gephyrin and collybistin, it anchors GABA_A_ receptors to the cytoskeleton of the postsynapse [23,24]. Other important neuronal anchoring proteins are LHFPL4, also known as GARLH4, interacting with γ2, α and β subunits [21,22] and radixin (Rdx), which is critical for extrasynaptic GABA_A_ receptor α5 cluster formation [25,26]. We performed single-cell RT-PCR to test the glial cells for expression of those molecules. All four anchoring proteins were found in developing hippocampal NG2 glia (Figure 7), with their relative expression frequencies remaining unchanged during development (Figure 7C). Since α5 increased in adulthood (Figure 6B), we also performed real-time RT-PCR with FAC-sorted hippocampal NG2 glia to test for potential regulation of radixin. However, radixin levels remained unchanged during development (Figure 7D). Thus, expression of the anchoring molecules collybistin, gephyrin, GARLH4 and radixin by hippocampal NG2 glia is not developmentally regulated. 

Next, we applied immunocytochemistry and expansion microscopy to detect anchoring molecules on the protein level. Triple staining against GFP/EYFP (for identification of NG2 glia), gephyrin (postsynaptic clusters) and radixin (extrasynaptic clusters) was performed in NG2-EYFPki mice at p60 (Figure 8). Z-stack evaluation and comparison with negative controls lacking first antibodies identified gephyrin- and radixin-positive puncta on glial membranes. Clusters of gephyrin and radixin puncta were observed on somata and processes of hippocampal NG2 glia. Gephyrin was also found in the cytoplasm. As in neurons [26], gephyrin and radixin puncta rarely overlapped indicating the presence of segregated clusters. Similar expression patterns of anchoring proteins were found along dendrites of Thy1-GFP-positive pyramidal cells (Figure 9), although in a few cases co-localization was obvious (Figure 9C).

## 3. Discussion

### 3.1. GABAergic Innervation of NG2 Glia Declines during Development

We report that GABAergic synaptic innervation of hippocampal NG2 glia decreased during postnatal maturation, indicated by reduced frequencies of spontaneous and miniature PSCs, which resembled developmental changes of the cells’ glutamatergic input [33]. With increasing age, the amplitudes of tonic currents of the glial cells remained similar, but the currents were more sensitive to antagonists of the α5-subunit, a component of extrasynaptic GABA_A_ receptors in neurons [26]. A similar developmental decrease in interneuron-NG2 glia synaptic signaling was observed in the barrel cortex [20], although in the adult hippocampus the sPSC frequency remained 10 times higher compared to the cortex (see also [16]). The former authors demonstrated that only at later stages of maturation, blocking GAT-1 led to increased ambient GABA, its spillover and slowed ePSC kinetics. In our study, rise and decay times of ePSCs were also slower in NG2 glia from older mice, suggesting altered subunit composition and/or preferred activation of extrasynaptic glial receptors in those cells. Prenatally, NG2 glia form functional clusters with GABAergic interneurons, and these cell clusters persist in the neonatal cortex and provide the basis for the close interactions of both cell types [34]. Thereafter, initial NG2 glia die and are substituted by newly born cells [35]. 

### 3.2. Molecular Changes of Glial GABA Receptors

Previously, we have shown that a majority of juvenile NG2 glia in the hippocampus express postsynaptic γ2-containing GABA_A_ receptors and that extrasynaptic receptors of these cells lacked γ2 [17]. In the developing barrel cortex, NG2 glia lose γ2 [19], an essential component of postsynaptic GABA_A_ receptors acting as a binding partner of the scaffold protein, gephyrin [36,37,38]. In the present study, downregulation of γ2 and synaptic signaling in the maturing hippocampus were accompanied by more frequent expression of α5 by NG2 glia. These molecular changes well corresponded to the functional alterations because (i) the tonic GABA receptor currents became more sensitive to an α5 subunit-specific antagonist and (ii) glial ePSC kinetics became slower, which is characteristic of α5-containing receptors [39,40]. The frequency of γ1 expression remained high in synaptic and extrasynaptic receptors of adult NG2 glia. 

Deletion of γ2 in juvenile cortical NG2 glia entailed lower frequencies and amplitudes of sPSCs, although glial proliferation, differentiation and Ca^2+^ signaling remained unaffected [41]. The density of NG2 glia decreases during development [42,43], but this decrease was even surpassed by γ2 deletion, indicating a role of this subunit in NG2 glia maintenance [41]. Thus, the physiological decline in NG2 glia density might be due to the cells’ developmental loss of γ2 and GABAergic synaptic input. In cortical slice cultures, the opposite was found, i.e., blocking of GABA_A_ receptors in oligodendrocyte lineage cells increased their numbers, suggesting that the synaptic input *decreases* numbers of NG2 glia and oligodendrocytes [44]. GABA receptor activation in NG2 glia in slice cultures leads to a downregulation of α1 and β2 subunits [44]. Other studies also suggested that differentiation and maintenance of NG2 glia are regulated by local GABAergic signaling [34,45]. In the juvenile neocortex, NG2 glia receive segregated synaptic input at proximal and distal sites from parvalbumin (PV)-positive, fast spiking and from PV-negative interneurons, respectively. Here, the proximal glial sites are equipped with γ2-containing receptors while the branches have receptors devoid of γ2 [46]. The innervation pattern in the juvenile hippocampus might be similar because in both regions NG2 glia share a comparable distribution and composition of GABA_A_ receptors [17,19]. Putative interaction partners of hippocampal NG2 glia are CCK-positive interneurons, which are located in the stratum radiatum, and also PV/SOM-positive bistratified cells in the stratum pyramidale [47,48].

### 3.3. Tonic GABA Receptor Currents

In CA1 pyramidal neurons, α5-containing GABA_A_ receptors are located extrasynaptically [49] to generate tonic currents [50,51,52,53]. These currents may reduce theta burst-induced LTP at CA3/CA1 synapses by preventing postsynaptic Ca^2+^ elevation and by increasing the threshold for LTP induction [54,55]. In NG2 glia, the amplitudes of tonic currents were the same in juvenile and adult mice, but only in the latter, part of the currents were mediated by α5 containing receptors. It is possible that in addition to neurons, astrocytes might have released GABA into the extracellular space and contributed to the tonic currents. Since juvenile NG2 glia have a very high input resistance, tonic GABA_A_ receptor activation and the resulting drop of the membrane resistance may fine-tune the efficiency of the excitatory GABAergic and glutamatergic synaptic input in these cells [17,33].

### 3.4. Auxiliary Subunits Und Anchoring Proteins

The GABAergic innervation of NG2 glia raised the question whether these synapses are similarly organized as neuronal inhibitory synapses. Typically, inhibitory synapses in neurons are formed by interactions of the γ2 subunit with gephyrin [36,38,56]. Another essential component of the postsynaptic receptor complex is collybistin that binds to neuroligin 2 (NLG2) to allow gephyrin-dependent GABA_A_ receptor clustering [23,24]. Throughout postnatal development, NG2 glia also express NLG2 [33], as well as gephyrin, collybistin and GARLH4 (present study). The latter is an auxiliary, regulatory subunit important for clustering of GABA_A_ receptor subunits with NLG2, e.g., in pyramidal cells of the hippocampus [21,22]. The γ2 subunit stabilizes GARLH4 expression in pyramidal cells. Despite downregulation of γ2 and loss of synaptic receptors during development, GARLH4 remained expressed in almost all NG2 glia of the adult hippocampus. 

The strong upregulation of the α5 subunit in NG2 glia with increasing age raised the question whether the cells also express radixin, which in neurons anchors α5-containing GABA_A_ receptors to extrasynaptic sites [26,57]. Indeed, radixin transcripts were found in the majority of the glial cells, both in juvenile and adult mice, and expansion microscopy confirmed segregated co-expression of radixin and gephyrin clusters at postsynaptic sites of NG2 glia. Thus, the expression pattern of gephyrin and radixin in NG2 glia resembled that observed in CA1 pyramidal neurons, suggesting that the proteins also have similar functions in the glial cells.

### 3.5. Impact of GABA Receptors in NG2 Glia

NG2 cells in the cerebellar white matter are innervated by local interneurons, and this synaptic input regulates proliferation and maturation of the glial cells. Disruption of GABAergic activity stimulated proliferation of NG2 glia and delayed their differentiation into oligodendrocytes. This process is critically dependent on the presence of the Na/K/2Cl (NKCC1)-transporter, which delivers an elevated intracellular Cl^−^ concentration and explains why the GABAergic input causes depolarization of the glial cells [58]. Deletion of the γ2 subunit in NG2 glia resulted in myelination deficits of the innervating PV+ fast spiking interneurons in the neocortex. As a consequence, the firing rate of these interneurons was reduced, as well as fast-forward inhibition of glutamatergic spiny stellate cells resulting in impaired whisker-based texture discrimination [59]. 

Due to the high intracellular Cl^−^ concentration of NG2, GABA receptor activation depolarizes the cells [16,17]. The same applies to immature neurons, where the transmitter was even shown to act as a trophic factor. Here, GABA-mediated depolarization activated voltage-gated Ca^2+^ channels and induced formation of postsynaptic gephyrin clusters and inhibitory synapses [60]. Because NG2 glia express a neuron-like set of GABA_A_ receptors and voltage-activated Ca^2+^ channels [17,32], and receptor activation mediates Ca^2+^ influx through these channels [16,32], it is conceivable that the GABAergic input serves similar functions in developing NG2 glia. Differentiation and proliferation of NG2 glia are critically dependent on neuronal activity [61]. Thus, the developmental decrease in synaptic innervation by GABAergic neurons could be a cause for the reduced proliferative potential of NG2 glia in the adult hippocampus [42]. Possibly, the GABAergic input to NG2 glia contributes to the release of neurotrophic factors from these cells and influences neuronal synaptic plasticity as demonstrated recently [43]. 

## 4. Materials and Methods

### 4.1. Animals

Experiments were performed with NG2-EYFPki mice [4] of both sexes that were either hetero- or homozygous for EYFP. These mice express EYFP under the control of the NG2 promotor allowing for reliable identification of NG2 cells by their fluorescent signal. Two developmental stages were compared, p7–10 (further on referred to as p10 group) and p54–75 (p60 group). Mice were kept under standard housing conditions in the animal facility of the Medical Faculty at University Bonn. All experiments were carried out in accordance with local, state and European regulations. 

### 4.2. Preparation of Acute Hippocampal Slices for Patch-Clamp Recordings and Cell Harvesting

Mice were anesthetized with isoflurane and decapitated. Brains were removed and cut with a vibratome (Leica VT1200S) into 300 µm thick horizontal sections in cold (4 °C) preparation solution containing the following (in mM): 1.25 NaH_2_PO_4_, 87 NaCl, 2.5 KCl, 7 MgCl_2_, 0.5 CaCl_2_, 25 glucose, 25 NaHCO_3_, and 61 sucrose (325–335 mOsm). After cutting, the slices were incubated for 15 min in preparation solution at 35 °C and afterwards transferred into artificial cerebrospinal fluid (aCSF) at 25 °C. aCSF consisted of the following (in mM): 1.25 NaH_2_PO_4_, 126 NaCl, 3 KCl, 2 MgSO_4_, 2 CaCl_2_, 10 glucose, and 26 NaHCO_3_ (305–315 mOsm). Both solutions were constantly oxygenated with carbogen (5% CO_2_/95% O_2_).

### 4.3. Electrophysiology

#### 4.3.1. Whole Cell Recordings

For patch-clamp recording, the slices were transferred into a recording chamber and constantly perfused with oxygenated aCSF. All experiments were conducted in the stratum radiatum of the CA1 region of the hippocampus at room temperature. An upright microscope (Nikon Eclipse E600FN, Nikon, Tokyo, Japan), equipped with infrared-DIC optics and epifluorescence, was used. NG2 glial cells were selected with a water-immersion 60× objective and identified by their EYFP fluorescence and their current pattern in the whole-cell configuration. Fluorescent somata enwrapping blood vessels (putative pericytes) were excluded. Cells were clamped at −80 mV. Recordings were obtained using an EPC9 amplifier (HEKA, Lambrecht, Germany), an EPC800amplifier (HEKA) or a combination of an EPC7 amplifier with an ITC16 AD/DA board (HEKA) and monitored using TIDA 5.24 software (HEKA). Patch pipettes were prepared from borosilicate capillaries (Science Products GmbH, Hofheim am Taunus, Germany) and had a resistance of 2–5 MΩ. For all experiments, except for cell harvesting (KCl-based), a CsCl-based intracellular solution was used, containing the following (in mM): 120 CsCl, 2 MgCl_2_, 0.5 CaCl_2_, 5 BAPTA, 10 HEPES, 3 Na_2_-ATP, and 10 TEA. Series and membrane resistance were monitored in constant intervals.

#### 4.3.2. Spontaneous and Miniature PSC

To record spontaneous postsynaptic currents (sPSC), a blocking solution containing 10 µM NBQX, 50 µM D-APV and 5 µM CGP55845 was applied via the perfusion system to isolate GABA_A_ receptor currents. After equilibration of the blocking solution in the recording chamber, spontaneous activity was recorded for 5 min. To record miniature postsynaptic currents (mPSC), 0.5 µM TTX was added to the blocking solution. mPSCs were recorded for 5–10 min. mPSCs were identified using template-based event detection (pClamp 10.0 software; Molecular Devices, San Jose, CA, USA). Events detected by template search were controlled visually before being accepted for further analysis, which was performed with Igor Pro 7 software (WaveMetrics, Lake Oswego OR, USA). Only recordings with at least two events were accepted for further calculations. Recordings were filtered at 1 kHz and sampled at 10 kHz.

#### 4.3.3. Evoked PSCs

Evoked postsynaptic currents (ePSCs) were characterized to investigate pharmacological properties of phasic GABA_A_ receptor currents. ePSCs were evoked by near-field electrical stimulation [9]. A monopolar stimulation electrode, consisting of a Teflon-coated silver wire with a chlorinated tip and a low resistance (<1 MΩ) aCSF-filled glass pipette, was placed close to the patch pipette. Stimuli were generated by a STG4004 stimulation device (Multichannel Systems, Reutlingen, Germany) in constant voltage mode. Cells were stimulated by applying a 100 µs biphasic single pulses every 15 s. At least 40 pulses were applied (~10 min) for each condition. To test for the presence of the γ2 subunit, diazepam (10 µM) and zolpidem (1 µM) ware added to the bath solution and its effects on ePSC amplitudes were tested. To isolate GABA_A_ receptor currents, the recordings were performed in the presence of 10 µM NBQX and 50 µM D-APV, blockers of AMPA and NMDA receptors. Series and membrane resistance were monitored in equal intervals. Recordings were filtered at 3 kHz and sampled at 10 kHz.

Using Igor Pro 7 software, peak amplitude, rise time (20–80%) and decay time were quantified. The effects of diazepam and zolpidem on receptor currents were determined as the ratios of the mean peak amplitudes in the presence of the drug and the mean peak amplitudes in control condition, individually for each cell recorded. 

#### 4.3.4. Tonic Currents

To detect tonic currents, 1 mM nipecotic acid, a GABA transporter blocker, was added to the bath solution for at least 3 min to elevate the extracellular GABA concentration. Tonic currents were then unmasked by adding 20 µM bicuculline, inducing a positive shift in the holding current due to blocking of GABA_A_ receptors. Only cells with stable holding currents (2 min) before application of nipecotic acid were included. To investigate pharmacological properties of tonic currents, the α5 subunit-specific blocker, α5IA, was used. First, 1 mM nipecotic acid was applied (at least 3 min to determine the baseline), and then, α5IA (500 nM) was added. Two min later, 20 µM of bicuculline was added to the bath solution. The effect of α5IA was then determined by measuring the shift of the holding current induced by application of α5IA and normalizing it to the shift of the holding current induced by bicuculline. In the following, this ratio is referred to as the relative effect of α5IA. Recordings were filtered at 1 kHz and sampled at 10 kHz.

### 4.4. Cell Harvesting and Single-Cell RT-PCR

For single-cell RT-PCR, cells were harvested with patch pipettes filled with KCl-based intracellular solution consisting of the following (in mM): 130 KCl, 2 MgCl_2_, 0.5 CaCl_2_, 5 BAPTA, 10 HEPES, and 3 Na_2_-ATP. After electrophysiological recording, the cytoplasm of individual cells was carefully aspirated into the patch pipette under microscopic control. The whole content of the patch pipette was then transferred into a reaction tube containing 3 µL DEPC-treated water, frozen in liquid nitrogen and stored at −20 °C until single-cell RT-PCR was performed.

Single-cell transcript analysis was performed as previously reported [62,63]. Reverse transcription (RT) was performed after addition of RT buffer (Invitrogen, Waltham, MA, USA), dNTPs (4 × 250 µM), random hexamer primers (50 µM; Roche, Basel, Switzerland), RNasin (20 U; Promega, Manheim, Germany) and Maxima H minus reverse transcriptase (100 U; Thermo Fisher Scientific, Waltham, MA, USA) at 37 °C for 1 h. For the multiplex two-round single-cell PCR, primers for α, β, γ, δ GABA_A_ receptor subunits and for PDGFRα were used (see [17,19]). To identify the α3 and auxiliary subunits as well as mRNAs for scaffolding proteins, a separate PCR together with primers for PDGFRα was performed (Table 1). Briefly, the first PCR was performed after adding PCR buffer, MgCl_2_ (2.5 mM), primers (200 nM each; 100 nM for PDGFRα), dNTPs (4 × 50 µM) and 5 U *Taq* polymerase (Invitrogen, Karlsruhe, Germany) to the RT product (final volume 50 µL). Thirty-five cycles were performed (denaturation at 94 °C, 25 s; annealing at 51 °C, 2 min for the first 5 cycles, and 45 s for the remaining cycles; extension at 72 °C, 25 s; final elongation at 72 °C, 7 min). An aliquot (2 µL) of the PCR product was used as template for the second PCR (35 cycles; annealing at 54 °C, first 5 cycles: 2 min, remaining cycles: 45 s) using nested primers. The conditions were the same as described for the first round, but Platinum *Taq* polymerase (2.5 U; Invitrogen) was added. Products were identified with gel electrophoresis using a molecular weight marker (low molecular weight marker, New England Biolabs, Frankfurt, Germany). As a positive control, RT-PCR for total RNA from mouse brain was run in parallel. Negative controls were performed using distilled water or bath solution for RT-PCR. The primers for the targets were located on different exons to prevent amplification of genomic DNA. To discriminate between β2 and β3 subunits, the PCR product was purified (MinElute PCR purification Kit, Qiagen, Hilden, Germany), and restriction analysis was performed (10 U enzyme, 6 h, 37 °C). PstI cut the PCR product of β2 subunit (307 bp) into 169 and 138 bp, while BanI cut the β3 product (307 bp) into fragments of 202 and 105 bp lengths.

### 4.5. FACsorting of NG2 Glial Cells and Semiquantitative Real-Time RT-PCR

Mice (NG2ki-EYFP, p10 and p60, male and female) were sacrificed, brains were dissected and the whole hippocampus was isolated under microscopic control (Stereo microscope, Zeiss, Germany). Cell suspension was prepared by mincing the tissue, digesting in papain at 37 °C for 15 min and incubating with DNAseI for another 10 min (Neural Dissociation Kit (P), Miltenyi, Bergisch-Gladbach, Germany). The tissue was dissociated using Pasteur pipettes and filtered through a 70 µm cell strainer. After addition of 10 mL HBSS (with Ca^2+^ and Mg^2+^) to the cell suspension, it was centrifuged at 300 g for 10 min. The pellet was suspended in 1 mL HBSS (without Ca^2+^ and Mg^2+^) and filtered through a 40 µm cell strainer. Fluorescent NG2 cells were identified using EYFP fluorescence emission at 527 nm and sorted using a FACSAriaIII flow cytometer (70 µm nozzle, BD Biosciences, Heidelberg, Germany) into tubes containing HBSS (without Ca^2+^ and Mg^2+^). After centrifugation at 2000 *g* for 10 min, the supernatant was discarded and the cells were suspended in 200 µL lysis/binding buffer (Invitrogen, Darmstadt, Germany), frozen in liquid nitrogen and stored at −80 °C.

Messenger RNA was isolated from isolated cells by cell lysis in the lysis/binding buffer and by using oligo(dT)25-linked Dynabeads (Invitrogen). Beads with the adherent mRNA were suspended in DEPC-treated water (20 µL). For first strand synthesis, first strand buffer (Invitrogen), dithiothreitol (DTT, 10 mM), dNTPs (4 × 250 µM, Applied Biosystems, Waltham, MA, USA), oligo-dT_24_-primer (5 µM, Eurogentec, Seraing, Belgium), RNasin (40 U, Promega, Madison, WI, USA) and SuperscriptIII reverse transcriptase (400 U, Invitrogen) were added, and the reaction mix was incubated for 1 h at 50 °C (final volume 40 µL). 

The reaction mixture for real-time PCR contained Takyon real-time PCR mastermix (Eurogentec, Seraing, Belgium) and Taqman primer/probe mix (Thermo Fisher Scientific, Darmstadt, Germany). One µL of the RT-product was added, and the reaction volume was 12.5 µL. PCRs for the respective target genes and β-actin, as housekeeping gene, were run in parallel wells for each sample, respectively, and triplicates for each sample were performed. Water served as negative control in each run. After denaturation (95 °C, 10 min), 50 cycles were performed (denaturation at 95 °C, 15 s; primer annealing and extension at 60 °C, 60 s; thermocycler CFX 384, Biorad, Munich, Germany). Fluorescence intensity was read out during each annealing/extension step. The target gene/β-actin gene expression ratio was determined by comparing C_T_ values of the target gene with those of the reference gene, β-actin. The relative quantification of different genes was determined according to 2^ΔΔCT^ method: X_target_/X_β-actin_ = 2^CTβ-actin − CTtarget^(1)
yielding a gene ratio with X being the input copy number and C_T_ the cycle number at threshold. By quantification of target gene expression against that of β-actin, ΔC_T_ was determined for each gene at the same fluorescence emission R_n_. 

### 4.6. Expansion Microscopy 

Mice were anesthetized by intraperitoneal injection of 70–90 µL of a xylazine (2%)/ketamine (100 mg/mL) mix in a ratio of 1:3. Afterwards, mice were transcardially perfused with 20–30 mL cold (4 °C) phosphate-buffered saline (PBS), followed by 20–30 mL 4% paraformaldehyde (PFA) in PBS. The brains were removed and stored overnight in 4% PFA at 4 °C, transferred into PBS the next day and stored for at least one day at 4 °C before slicing. Brains were cut with a vibratome (Leica VT1200S) in cold (4 °C) PBS into 70 µm thick coronal slices. Slices were stored at 4 °C in PBS containing sodium azide until staining.

The expansion microscopy (ExM) protocol for imaging proteins with conventional primary and secondary antibodies was adopted from [64,65,66]. Fixed coronal hippocampal slices (70 µm thick) were blocked overnight at 4 °C in blocking buffer (5% normal goat serum, 1% Triton X-100 in PBS pH 7.4). Primary antibodies were incubated in blocking buffer for 48 h at 4 °C. Antibodies used were chicken anti-GFP (1:2000; ab13970, Abcam (Cambridge, UK) lot: GR236651-g), rabbit anti-Radixin (1:200; PA5-21660, Invitrogen), and mouse anti-Gephyrin (1:200; 147 021, Synaptic Systems, Göttingen, Germany). After washing (PBS, 3 × 20 min, room temperature), secondary antibodies were incubated overnight at 4 °C in blocking buffer. Secondary antibodies used were goat anti-chicken Alexa Fluor 488 (1:200; A11039, ThermoFisher lot: 1691381, Waltham, MA, USA), goat anti-mouse Alexa Fluor 568 (1:200; A1131, ThermoFisher), and goat anti-rabbit biotin (1:200; 111-066-144, Jackson ImmunoResearch, West Grove, PA, USA). The following day, after washing (PBS, 3 × 20 min, RT), Hoechst 33342 was incubated for 10 min at room temperature in distilled water (1:2000). After washing (PBS, 3 × 20 min, room temperature), the slices were pre-imaged with a 40×/1.1 NA objective in a Leica SP8 confocal microscope and a z-stack (1 µm interval) of the tip of dentate gyrus (DG) was acquired for expansion factor calculation (only Hoechst 33342 signal imaged). After staining, the slices were incubated with the linker molecule methylacrylic acid-NHS (1 mM) for 1 h at room temperature. After washing (PBS, 3 × 20 min, room temperature), slices were incubated for 45 min at 4 °C in monomer solution (in g/100 mL PBS): 8.6 sodium acrylate, 2.5 acrylamide, 0.15 N,N’-methylenebisacrylamide, and 11.7 NaCl. Then, slices were incubated for 5 min at 4 °C in gelling solution (monomer solution supplemented with % (*w*/*v*): 0.01 4-hydroxy-TEMPO; 0.2 TEMED; 0.2 ammonium persulfate). Slices were placed in the gelling solution on a glass slide and the preparation was covered with a coverslip and incubated for 2 h at 37 °C. Coverslips and excess gel around the slice were removed, and gels incubated overnight at 25 °C in digestion buffer containing 50 mM Tris, 1 mM EDTA, 0.5% Triton-X100, 0.8 M guanidine hydrochloride, 16 U/mL of proteinase K, pH 8.0. The following day, the digestion buffer was removed, and the gels were incubated with streptavidin Alexa Fluor 647 (1:200; 016–600-084, Jackson ImmunoResearch lot: 124695) in PBS for 2 h at room temperature. For expansion, slices were then incubated in distilled water (pH 7.4 adjusted with NaOH) for 2.5 h, at room temperature, and water was exchanged repeatedly every 15–20 min. Finally, slices were mounted on poly-lysine coated µ-Slide 2 well Ibidi-chambers and sealed with a poly-lysine coated coverslip on top, adding a drop of water to prevent gel drying. Ibidi chambers and coverslips were coated by incubating with poly-L-lysine solution (0.01% *w*/*v* in water, P8920, Sigma-Aldrich lot: 050M4339, St. Louis, MO, USA) for at least 45 min at room temperature by shaking, and dried then with pressured air. Imaging of regions of interest was performed on a Leica SP8 inverted confocal microscope using a 40×/1.1 NA objective and hybrid detectors. Images were deconvoluted in Leica Systems software LASAF 3.3 and processed with Fiji 1.53. Prior to imaging the region of interest for each slice, the tip of the DG was imaged with a 20×/0.75 NA objective (z-stack 6 µm interval with Hoechst 33342 signal only). The expansion factor was calculated by identifying the same cells labeled with Hoechst 33342 in the tip of DG before and after expansion and averaging the expansion factor of 10 measures.

### 4.7. Data Analysis and Statistics 

Data analysis was performed using Igor Pro 7 and pClamp 10.0 software. Statistical analysis was carried out with R 3.5.0 software. All data were tested for normal distribution using Shapiro–Wilk test. Normally distributed data were tested for homogeneity of variance using Levene’s test and differences in data were tested for significance using Student’s *t*-test either with or without Welch correction accordingly. Differences in not normally distributed data were analyzed with Mann–Whitney-U test. Data from single-cell RT-PCR was tested for significant differences using Chi-square test. Level for significance was set as *p* < 0.05. Data are shown as bar plots with mean ± SD or, in case of not normally distributed data, as Tukey’s box plots with median, quartiles (25% and 75%) and whiskers (±1.5 times the interquartile range). Since no sex-specific differences were found in our analyses, data from male and female mice have been pooled.

## Figures and Tables

**Figure 1 ijms-24-13490-f001:**
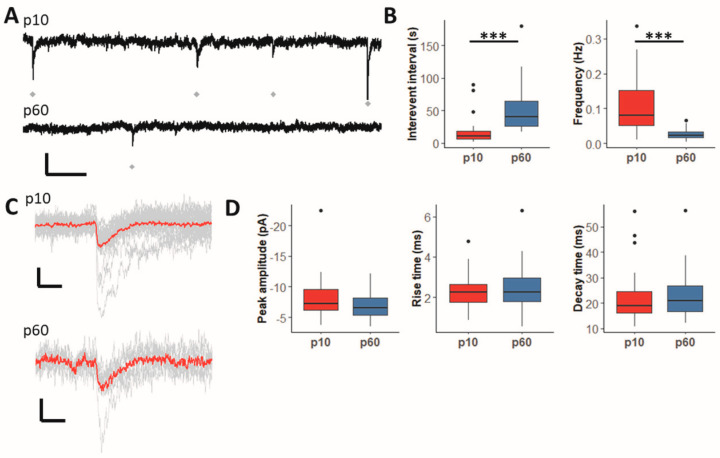
The frequency of spontaneous GABAergic PSCs is downregulated during postnatal development. (**A**) Original recordings of sPSCs in NG2 glial cells of juvenile (upper trace) and adult (lower trace) mice. Recordings were performed in the presence of 10 µM NBQX, 50 µM D-APV and 5 µM CGP55845 to isolate GABAergic currents. Scale bars, 20 pA and 1 s. Gray diamonds indicate events. (**B**) Boxplots showing interevent interval and frequency of sPSCs for all recorded cells. The frequency was significantly lower at age p60 than at age p10 (p60: 0.0223 Hz, p10: 0.0795 Hz, *p* < 0.001). Thus, the interevent interval was significantly higher in adult mice than in juvenile mice (p60: 40.77 s, p10: 10.59 s, *p* < 0.001). (**C**) Recorded events (gray lines) and their average (red line) for a single NG2 cell at ages p10 (upper trace) and p60 (lower trace). Scale bars show 5 pA and 20 ms. (**D**) Boxplots showing peak amplitude, rise time (defined as rise time from 20% of peak amplitude to 80% of peak amplitude) and decay time (defined as the decay time constant tau). There was no significant difference between p10 and p60 for those parameters (amplitudes, p10: −7.28 pA, −9.54 pA–−6.20 pA; p60: −6.51 pA, −8.20 pA to −5.34 pA; *p* = 0.079; rise time, p10: 2.25 ms, 1.77–2.63 ms; p60: 2.25 ms, 1.81–2.98 ms; *p* = 0.45; decay time, p10: 18.93 ms, 16.15–24.58 ms; p60: 21.02 ms, 26.5–65.2 ms; *p* = 0.72). Mann–Whitney-U test, *** *p* < 0.001. p10: n = 34, N = 13; p60: n = 31, N = 11.

**Figure 2 ijms-24-13490-f002:**
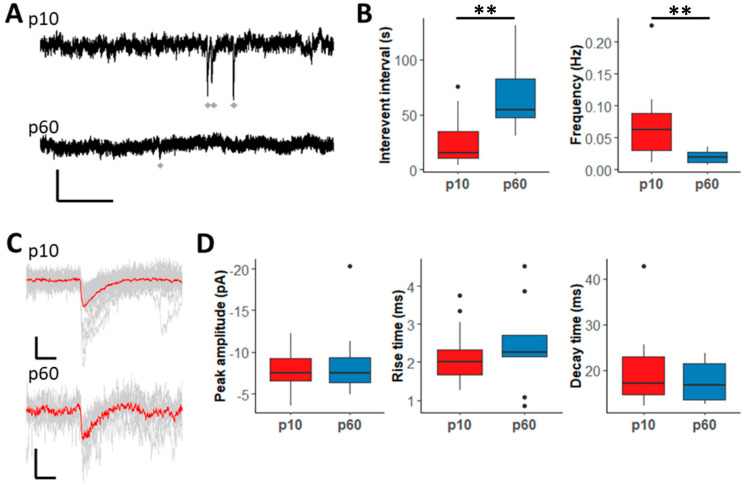
The frequency of GABAergic miniature PSCs in hippocampal NG2 cells is lower in adult animals. (**A**) Original recordings of mPSCs in NG2 cells of juvenile (upper trace) and adult (lower trace) animals. For recording, 0.5 µM TTX was added to the blocking solution (10 µM NBQX, 50 µM D-APV and 5 µM CGP55845). Gray diamonds indicate events. In p60 mice only few mPSCs could be observed. Scale bar shows 10 pA and 1 s. (**B**) Boxplots showing interevent interval and frequency of mPSCs for all recorded cells. At age p60, the interevent interval was significantly higher (p60: 54.02 s, p10: 15.28 s, *p* < 0.01) and the frequency was significantly lower (p60: 0.0191 Hz, p10: 0.063 Hz, *p* < 0.01) than at age p10. (**C**) Recorded mPSCs (gray lines) and their average (red line) for a single NG2 cell at ages p10 (upper trace) and p60 (lower trace). Scale bar shows 5 pA and 20 ms. (**D**) Boxplots showing peak amplitude (p10: −7.54 pA; p60:–7.46 pA, *p* = 0.76), rise time (p10: 2.01 ms; p60: 2.27 ms; *p* = 0.59) and decay time of mPSCs (p10: 17.16 ms; p60: 16.73 ms; *p* = 0.39) did not differ between juvenile and adult mice. Mann–Whitney-U test, ** *p* < 0.01. p10: n = 20, N = 9; p60: n = 10, N = 9.

**Figure 3 ijms-24-13490-f003:**
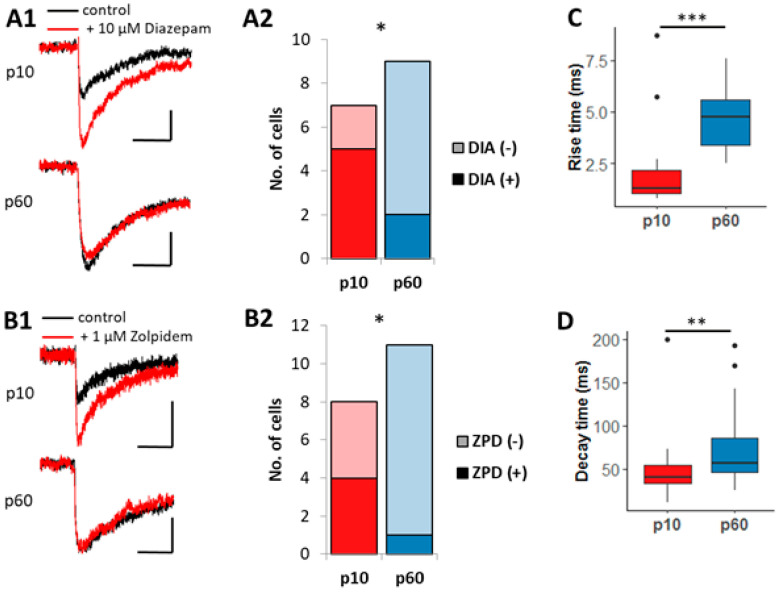
Evoked GABAergic postsynaptic currents in NG2 glia of juvenile mice are more sensitive to the γ2 subunit selective modulators diazepam and zolpidem. GABAergic ePSCs were evoked by electrical near-field stimulation and recorded in the presence of 10 µM NBQX and 50 µM D-APV. (**A1**,**B1**) Averaged ePSCs of a single NG2 cell under control conditions (black traces) and after applying 10 µM diazepam (**A1**, red trace) or 1 µM zolpidem (**B1**, red trace) at p10 (upper traces) and p60 (lower traces). Scale bars: 4 pA, 50 ms; stimulation artefacts were blanked. In sensitive cells, diazepam and zolpidem increased ePSC amplitudes. (**A2**) Plot comparing the number of diazepam-sensitive cells (DIA (+), dark color) and insensitive cells (DIA (−), light color) at ages p10 and p60. At age p10, more cells were sensitive to diazepam (p10: 5/7 cells; p60: 2/9 cells; *p* < 0.05). (**B2**) Plot comparing the number of zolpidem-sensitive cells (ZPD (+), dark color) and non-sensitive cells (ZPD (−), light color) in juvenile and adult mice. At the juvenile stage, more cells were sensitive to zolpidem (p10: 4/8 cells; p60: 1/11 cells; *p* < 0.05). Chi-square test; * *p* < 0.05. (**C**,**D**) Boxplots comparing rise (**C**) and decay times (**D**) of ePSCs in control conditions between p10 and p60. Both parameters (rise time p10: 1.28 ms, n = 17, N = 8; p60: 4.74 ms, n = 11, N = 7; *p* < 0.001; decay time p10: 40.56 ms, n = 23, N = 8; p60: 57.36 ms, n = 21, N = 7; *p* < 0.01) were slower in adult mice. Rise time was only determined if the stimulus artefact did not overlay with the event. Mann–Whitney-U test, ** *p* < 0.01, *** *p* < 0.001.

**Figure 4 ijms-24-13490-f004:**
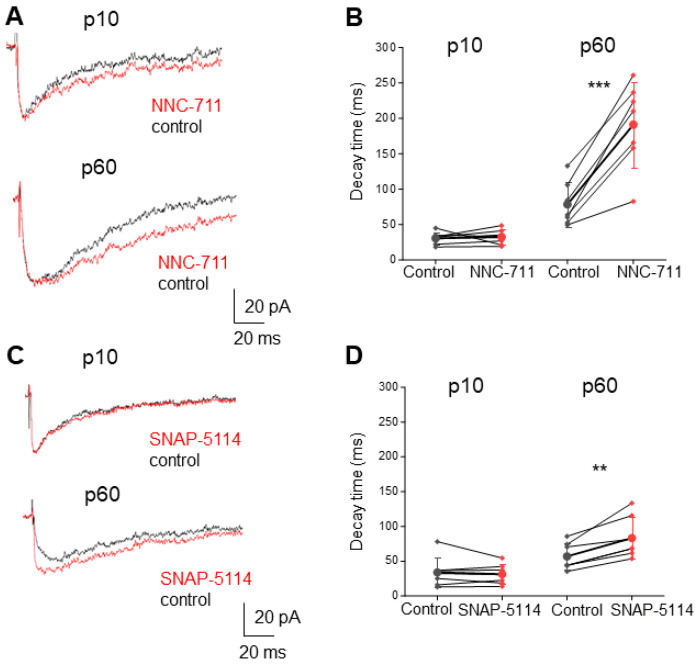
GABA uptake blockers prolong ePSC decay only in adult NG2 glia. (**A**,**C**) Original recordings of ePSCs in NG2 glia at p10 and p60 before (black trace) and after application (red) of NNC-711 (**A**, 10 µM) and SNAP-5114 (**C**, 100 µM). Bath solution contained 10 µM NBQX, 50 µM D-APV and 5 µM CGP55845. (**B**,**D**) Plots comparing changes in decay time constant (mono-exponential fit) after application of NNC-711 (**B**; p10 control: 30.52 ± 8.72 ms; NNC-711: 32.22 ± 10.66 ms, n = 7, N = 5; *p* = 0.73; p60 control: 78.42 ± 31.14 ms; NNC-711: 190.96 ± 60.32 ms, n = 7, N = 4; *p* < 0.001) and SNAP-5114 (**D**; p10: control: 33.95 ± 21.63 ms; SNAP-5114: 31.35 ± 14.41 ms, n = 7; N = 4; *p* = 0.53; p60 control: 56.83 ± 19.45 ms; SNAP-5114: 83.18 ± 30 ms, N = 3, n = 7; *p* < 0.01). Thus, only at the adult stage, inhibition of GABA uptake slowed ePSC decay. Paired *t*-test; ** *p* < 0.01, *** *p* < 0.001.

**Figure 5 ijms-24-13490-f005:**
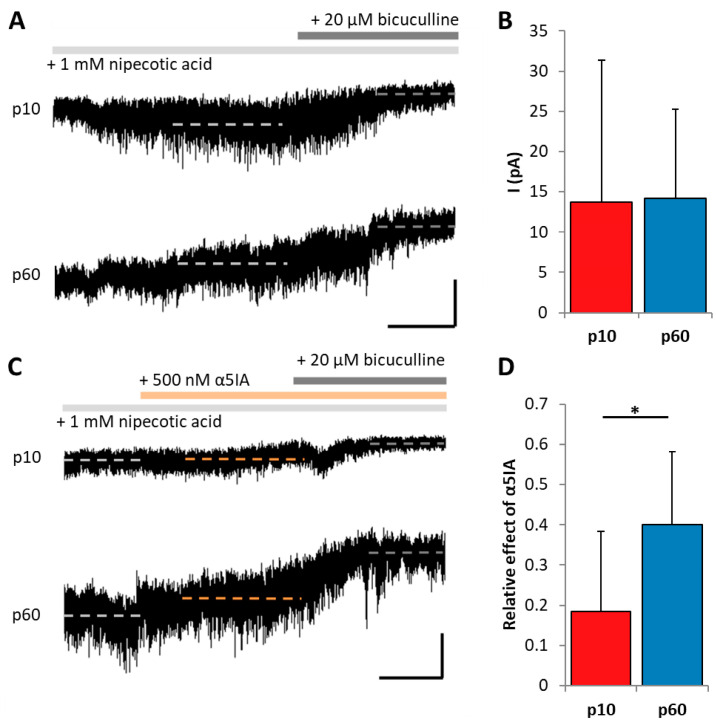
Tonic GABA currents of adult NG2 glia show higher sensitivity to α5 subunit blockage. (**A**) Original recordings of tonic currents in NG2 glis at p10 (upper trace) and p60 (bottom). The trace was smoothened by a boxcar averaging algorithm. Nipecotic acid (1 mM) was added to the bath solution for at least 3 min to increase extracellular GABA levels. Subsequent addition of bicuculline (20 µM) induced a positive shift of holding current. The dashed line indicates the average of the labeled range in the presence of nipocotid acid (light gray) and nipecotic acid/bicuculline (dark gray), respectively. Scale bar: 10 pA, 50 s. (**B**) Bar graph showing the mean shift of holding current (p10: 13.72 ± 17.61 pA, n = 17, N = 5; p60: 14.15 ± 11.14 pA, n = 14, N = 4). There was no difference between ages (*t*-test; *p* = 0.09). (**C**) Original recording demonstrating the effect of the α5 subunit specific blocker, α5IA, in NG2 glia of juvenile (upper trace) and adult mice (bottom). Nipecotic acid was applied as described in (**A**). Traces were smoothened as mentioned in (**A**). Subsequent application of α5IA (500 nM) partially blocking tonic currents. After another 2 min, bicuculline (20 µM) was added to the bath solution to block the remaining tonic current. Dashed line indicates the average of the labeled range in the presence of nipocotid acid (light gray), after addition of α5IA (orange) and bicuculline (dark gray), respectively. Scale bar: 10 pA, 50 s. (**D**) Bar graph showing the relative effect of α5IA on GABAergic tonic current. The shift in holding current induced by α5IA was normalized to that induced by bicuculline. The relative effect of α5IA was higher in p60 mice (p10: 0.18 ± 0.2; n = 9, N = 5; p60: 0.4 ± 0.18, n = 9, N = 4). Two-sample *t*-test, * *p* < 0.05.

**Figure 6 ijms-24-13490-f006:**
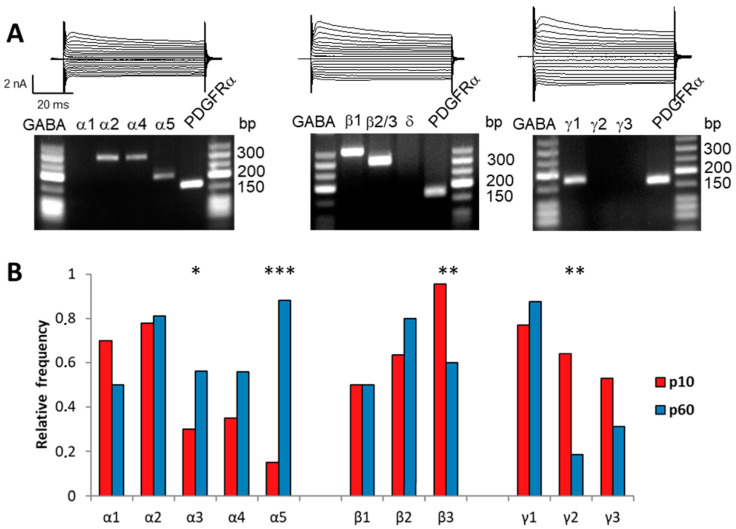
Subunit composition of GABA_A_ receptor in NG2 glia changes during development. (**A**) Single-cell RT-PCR was performed to analyze the expression of GABA_A_ receptor subunits by NG2 cells in the CA1 stratum radiatum at p60. Current patterns (voltage steps between −160 mV and +20 mV, increment 10 mV, holding potential −80 mV) of individual cells are shown together with the respective agarose gels of PCR products for α, β, γ and δ subunits. PDGFRα served as positive control. (**B**) Relative expression frequency of α, β and γ subunits at p60 (blue bars) compared to the juvenile stage (red; data obtained from [17]). β2 and β3 subunits were differentiated using restriction analysis. The frequencies of α5 and α3 expression were strongly upregulated at p60 (p10: 0.15 and 0.3 vs. p60: 0.88 and 0.56; *p* < 0.001 and *p* < 0.05, respectively). In contrast, β3 and γ2 transcripts were less frequent in cells from adult mice (p10: 0.95 and 0.64 vs. p60: 0.6 and 0.19; *p* < 0.01 each). Chi-square test; * *p* < 0.05; ** *p* < 0.01; *** *p* < 0.001.

**Figure 7 ijms-24-13490-f007:**
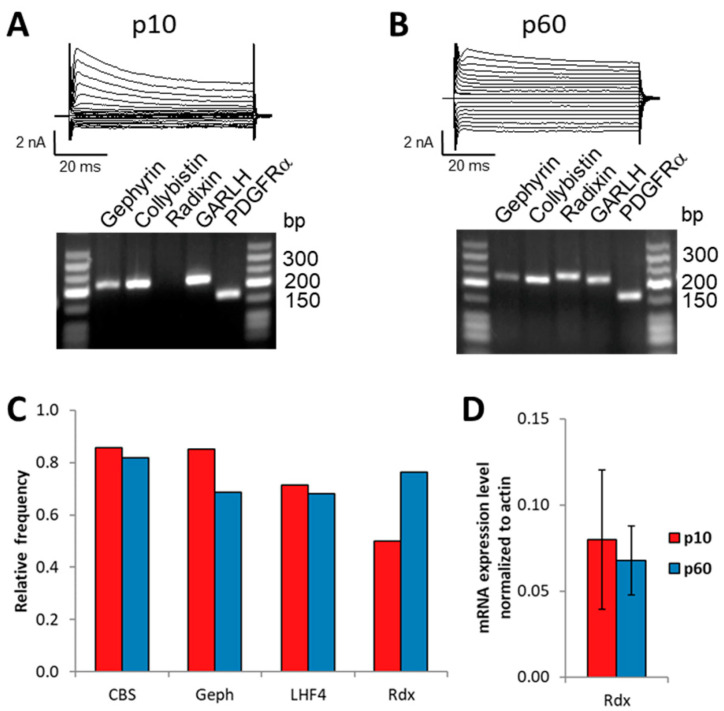
Hippocampal NG2 glia express transcripts for postsynaptic and extrasynaptic GABA_A_ receptor clustering proteins and auxiliary receptor subunits. (**A**,**B**) Single-cell RT-PCR of NG2 glia was performed to detect GABA_A_ receptor anchoring proteins. Current patterns (voltage steps between −160 mV and +20 mV, 10 mV increment, holding potential −80 mV) are shown together with the respective agarose gel of PCR products for collybistin (CBS), gephyrin (Geph), radixin (Rdx) and GARLH4 (LHF4) at p10 (**A**) and p60 (**B**). PDGFRα served as positive control. (**C**) Single-cell RT-PCR analysis of different GABA_A_ receptor associated anchoring proteins at p10 (0.86, n = 21; 0.85, n = 20; 0.71, n = 21; 0.5, n = 18) and p60 (0.82, n = 22; 0.69, n = 16; 0.68, n = 22; 0.76, n = 17). There were no age-dependent differences (*p* > 0.1 in all cases, Chi-square test). (**D**) Real-time RT-PCR analysis for radixin performed on FAC-sorted NG2 glia from the hippocampus at p10 (normalized values; 0.08 ± 0.04, n = 15) and p60 (0.068 ± 0.02, n = 13). No age-dependent differences were found (*t*-test, *p* = 0.33).

**Figure 8 ijms-24-13490-f008:**
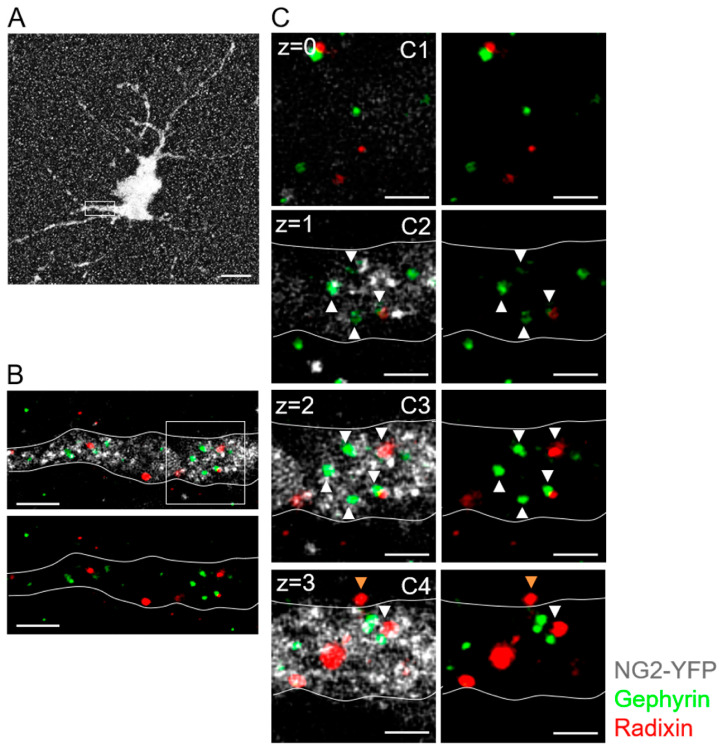
Expansion microscopy reveals gephyrin and radixin expression in processes of NG2 glia. (**A**) A single EYFP-positive NG2 glial cell in maximum z-stack projection is displayed after antibody staining against GFP (white). (**B**) Antibody staining against gephyrin (green) and radixin (red) in the boxed detail from (**A**) showed accumulation of gephyrin and radixin in a NG2 glial process (GFP, white). The lower panel displays gephyrin (green) and radixin (red) puncta in the cytoplasm and on the membrane without the GFP signal. Gephyrin and radixin mostly localized in separate clusters. (**C**) Inset box from (**B**) at higher magnification. Primary glial process showed localization of gephyrin and radixin in the cytoplasm (white arrowhead) or contacting the membrane surface (orange arrowhead) in consecutive z planes. Scale bar: (**A**): 5 µm; (**B**): 1 µm; (**C**): 500 nm. Z intervals = 2 μm post-expansion, 435 ± 39 nm pre-expansion.

**Figure 9 ijms-24-13490-f009:**
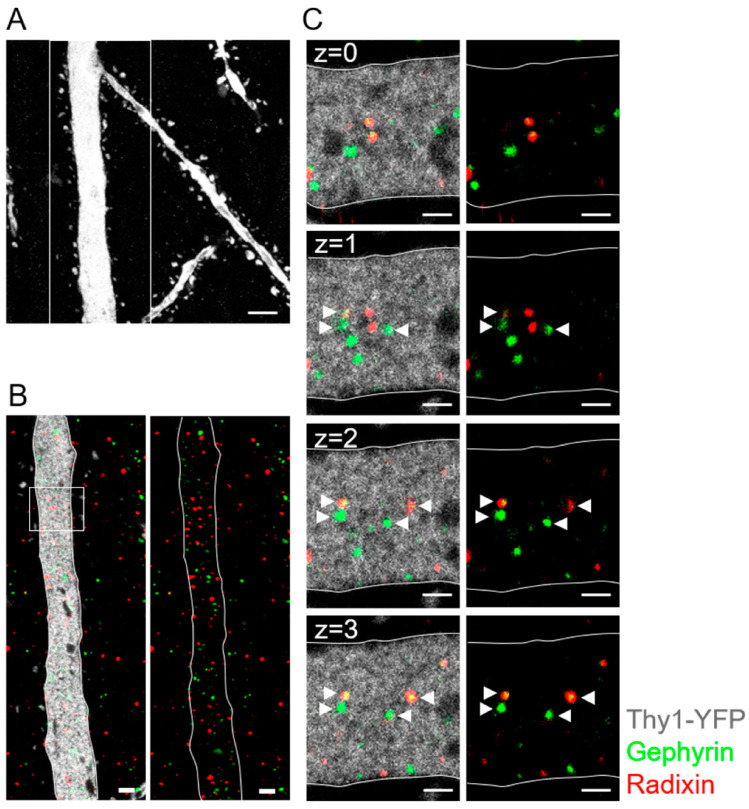
Expansion microscopy reveals gephyrin and radixin expression in dendrites of CA1 pyramidal neurons. (**A**) A single EYFP-positive dendrite of a CA1 pyramidal neuron in maximum z-stack projection is displayed after antibody staining against GFP (white). (**B**) Antibody staining against gephyrin (green) and radixin (red) in the boxed detail from (**A**) showed accumulation of gephyrin and radixin in the neuronal dendrite (GFP, white; single optical plane). The right panel displays gephyrin (green) and radixin (red) puncta in the cytoplasm of dendrite without the GFP signal. (**C**) Inset box from (**B**) at higher magnification. The primary dendrite showed localization of gephyrin and radixin in the cytoplasm (white arrowhead) in consecutive z planes. Note: co-localization of gephyrin and radixin in some clusters. Scale bar: (**A**): 2.5 µm; (**B**): 1 µm; (**C**): 500 nm. Z intervals = 0.5 μm post-expansion, 124 ± 7 nm pre-expansion.

**Table 1 ijms-24-13490-t001:** Primers used for single-cell RT-PCR.

Gene	Primer Sequence	Product Length	Position	Genbank Accession Number
Collybistin (*Arhgef9*)	se 5′-CACGGAGCGCCATTACATCAA as 5′-GGCGGCAGGCCTCAAAGA	338 bp	339 659	NM_001033329
Collybistin (*Arhgef9*) (nested)	se 5′-GTGCCGAAAGAGAAGGGACAT as 5′-CAGGCATCCAGGTGGTTGTTA	219 bp	396 594	
*Gephyrin*	se 5′-AGGTGCAGCAGCAAGGAGAACATT as 5′-GTTGTAACCCGCATCACTTGTCCT	365 bp	982 1323	NM_145965
*Gephyrin* (nested)	se 5′-TCTCCTTTTCCCCTGACG as 5′-TCCCCAATGATGAAACGAT	221 bp	1063 1265	
*Radixin*	se 5′-CAGCGAGCAAAAGAGGAGGCAGAG as 5′-TGCTTCTTCACGCGTTCATTCTTC	443 bp	1138 1557	NM_009041
*Radixin* (nested)	se 5′-GCAAGCTGCTGACCAGATGAAG as 5′-TCCGTGGGAGGGATGACTG	234 bp	1215 1430	
*LHFPL4*	se 5′-TCTACAAGATCTGCGCCTGGATGC as 5′-TGGCCGCAGCACGGAACTTACTGT	311 bp	371 658	NM_177763
*LHFPL4* (nested)	se 5′-ATGGCTGGGATGCTGAGACC as 5′-ACTGTAGTGCCCACGAAATCTTTG	220 bp	443 639	
GABA_A_-R α3 (*Gabra3*)	se 5′-TAACCGGCTTCGACCTGGACTTG as 5′-CAGCGTGTATTGTTAACCTCATTG	332 bp	237 545	NM_008067
GABA_A_-R α3 (*Gabra3*) (nested)	se 5′-AGTTTTGGCCCTGTGTCAGAC as 5′-AGAGGAGGGTCCCATTATCTACC	241 bp	301 519	

Position 1 is the first nucleotide of the initiation codon. The length of PCR products was indicated as base pairs (bp). ‘se’ and ‘as’ indicate sense and antisense primers. All sense and antisense primers are located on different exons, respectively.

## Data Availability

The data presented in this study are available in the article and on request from the corresponding authors.

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
