# Peer review of "Impact of Developmental Changes of GABAA Receptors on Interneuron-NG2 Glia Transmission in the Hippocampus"

_ijms, 2023, doi:10.3390/ijms241713490_

Round 1
Reviewer 1 Report
In the current manuscript, Patt L and co-workers investigated the developmental changes of interneuron-NG2 glia transmission in the hippocampus. The overall study is interesting and well design. Authors have used electrophysiological and molecular approaches to address interneuron-NG2 glia transmission.
There are only a few comments as follows:
1) Line 260: Please mention the author name for the cited reference.
2) Line 380, 383, and 541 typo’s.
3) Although the author mentioned that they had used both sexes in the current study, therefore please add a relevant paragraph of discussion on the results.
4) If the final analysis is done while considering both the sex then a dot plot with color coding for male and female data sets will further help to understand the importance of results.
Author Response
Response to the Reviewers (changes labelled)
Reviewer 1
1) Line 260: Please mention the author name for the cited reference.
Response: Thanks, in the revision we give the first author of the citation as suggested.
2) Line 380, 383, and 541 typo’s
Response: We have corrected the typo in line 380, no typos were found in lines 383 and 541.
3) Although the author mentioned that they had used both sexes in the current study, therefore please add a relevant paragraph of discussion on the results.
Response: We have carefully tested the results for sex differences, but didn’t find any. We have added this information to the Methods section of the revised manuscript (lines 666/667).
4) If the final analysis is done while considering both the sex then a dot plot with color coding for male and female data sets will further help to understand the importance of results.
Response: In the p10 pups, sex was not determined, while no sex-related differences were found at p60. Therefore, we do not believe it is appropriate to present gender-specific data in the figures. As an illustration for the reviewer, we include here an example figure showing the distribution of sex-specific data points. We think it makes more sense to present the data as box plots and would like to stick to our previous presentation.
Figure for Reviewer 1

Reviewer 2 Report
The submitted manuscript is devoted to studying aspects of GABA(A)R-mediated signaling in such a mysterious population of brain cells as NG2-glia. The manuscript can be recommended for publication after the revision.
- Title. Interneuron-NG2-glia interaction was only implied but was not directly demonstrated. The authors have shown the critical role of the extrasynaptic GABA(A)Rs which can also be activated by the astrocyte-released GABA.
- The Introduction needs a brief description of the functional role, the percentage, markers of NG2-positive cells. This information is critical for understanding the results and the scientific problem in general.
- Figure 5. Which current is tonic in Fig. 5A? Please, show the averaged smoothed traces in addition to the presented recordings. I have found that a positive shift of the holding current is observed even without bicuculline in p60 trace.
- The Discussion should be revised, and the summary/conclusion has to be included.
- Some researchers call NG2-glia "NG2-positive cells," highlighting the possible non-glial (neuronal) fate of these cells. The homogeneity of NG2-positive cells is debatable. In this regard, can the changes in GABA(A)R-mediated signaling be explained by the developmental changes in the heterogeneity of NG2-cells? Please, discuss it.
- The text requires revision since numerous typos are present ("anesthetics (reviewed by [4-6].", "of glia GABAA receptor", "[32] reported that NG2 glia" etc.).
The typos have to be corrected.
Author Response
Response to Reviewers
Reviewer 2
1) Title. Interneuron-NG2-glia interaction was only implied but was not directly demonstrated. The authors have shown the critical role of the extrasynaptic GABA(A)Rs which can also be activated by the astrocyte-released GABA.
Response: Previous work, e.g. by Bergles’ group, our own (e.g. Jabs et al., 2005; Haberlandt et al., 2011) and others, has clearly demonstrated direct monosynaptic interneuron-NG2 glia synaptic interactions. In line with these previous findings, the present study reports spontaneous (Figs. 1, 2) and single stimulus-correlated responses of NG2 glia (Fig. 3) with a short delay and fast kinetics, strongly suggesting that the glial responses were due to vesicular GABA release from interneurons. Thus, we think the title is justified. For analysis of extrasynaptic responses we also used very short, single-pulse electrical stimulation and observed rapid responses and decay kinetics in the ms range (Fig. 4), again strongly indicating that synaptic release of GABA from neurons and its spillover was detected by NG2 glia. Actually, much stronger electrical stimulation is required to provoke transmitter release from astrocytes, which occurs with a much slower delay, see e.g. Serrano et al., 2006, J Neurosci. Since the experiments were conducted with the GABA transporters being blocked, GABA release from astrocytes through inverted transporters can be excluded.
We agree with the Reviewer that astrocytic GABA release might have contributed to the tonic currents described in Fig. 5, and we have modified the text accordingly (line 217/218).
2) The Introduction needs a brief description of the functional role, the percentage, markers of NG2-positive cells. This information is critical for understanding the results and the scientific problem in general.
Response: We have added this information to the revised Introduction as requested.
3) Figure 5. Which current is tonic in Fig. 5A? Please, show the averaged smoothed traces in addition to the presented recordings. I have found that a positive shift of the holding current is observed even without bicuculline in p60 trace.
Response: In the revised Fig. 5, we give now smoothened traces as suggested by the Reviewer (boxcar averaging algorithm). In addition, dashed lines help now to better recognize the changes in tonic currents.
4) The Discussion should be revised, and the summary/conclusion has to be included.
Response: We have revised the Discussion, adding the points as discussed above. A conclusion is optional; our conclusion are summarized in the final paragraph ‚Functional implications‘ at the end of Discussion.
5) Some researchers call NG2-glia "NG2-positive cells," highlighting the possible non-glial (neuronal) fate of these cells. The homogeneity of NG2-positive cells is debatable. In this regard, can the changes in GABA(A)R-mediated signaling be explained by the developmental changes in the heterogeneity of NG2-cells? Please, discuss it.
Response: We prefer the term ‘NG2 glia’ to distinguish these cells from pericytes, which are also NG2 positive. In the present study we used NG2 knockin EYFP mice, where all cells with NG2 promoter activity are fluorescent. These fluorescent cells express markers typical of NG2 glia (NG2, PDGFRa) and oligodendrocyte progenitor cells (O4, Olig2, Sox10) while no co-expression of markers for neurons, astrocytes and microglia was found (Karram et al., 2008). In our study, cells contacting blood vessels were excluded. We included this information in the revised Introduction and Methods. It is widely agreed in the field that under physiological conditions NG2 positive cells do not differentiate into neurons (e.g. Dimou et al., 2008, J Neurosci; Kang et al., 2010, Neuron; Huang et al., 2019, Glia)
It is well known that several properties of NG2 glia change during development and also partly depend on the genetic background (see e.g. Kressin et al., 1996, Glia; Moshrefi-Ravasdjani et al., 2017). The present study adds to this insight by demonstrating that GABAA receptor expression and neuron-NG2 glia signaling in the hippocampus also change during postnatal maturation. Whether and to which extent there is heterogeneity in GABA signaling at a given developmental stage has never been investigated. From our data, there is no evidence that changes in the heterogeneity of NG2 glia occur during development.
6) - The text requires revision since numerous typos are present ("anesthetics (reviewed by [4-6].", "of glia GABAA receptor", "[32] reported that NG2 glia" etc.).
Response: We have corrected the typos.
Round 2
Reviewer 2 Report
The authors have addressed all my comments.